# Intracoronary Injection of Autologous CD34+ Cells Improves One-Year Left Ventricular Systolic Function in Patients with Diffuse Coronary Artery Disease and Preserved Cardiac Performance—A Randomized, Open-Label, Controlled Phase II Clinical Trial

**DOI:** 10.3390/jcm9041043

**Published:** 2020-04-07

**Authors:** Pei-Hsun Sung, Yi-Chen Li, Mel S. Lee, Hao-Yi Hsiao, Ming-Chun Ma, Sung-Nan Pei, Hsin-Ju Chiang, Fan-Yen Lee, Hon-Kan Yip

**Affiliations:** 1Division of Cardiology, Department of Internal Medicine, Kaohsiung Chang Gung Memorial Hospital and Chang Gung University, College of Medicine, Kaohsiung 83301, Taiwan; e12281@cgmh.org.tw (P.-H.S.); ryichenli@gmail.com (Y.-C.L.); haoyi222@cgmh.org.tw (H.-Y.H.); 2Center for Shockwave Medicine and Tissue Engineering, Kaohsiung Chang Gung Memorial Hospital, Kaohsiung 83301, Taiwan; 3Department of Orthopedics, Kaohsiung Chang Gung Memorial Hospital and Chang Gung University, College of Medicine, Kaohsiung 83301, Taiwan.; mellee@cgmh.org.tw; 4Division of Hema-Oncology, Department of Internal Medicine, Kaohsiung Chang Gung Memorial Hospital and Chang Gung University College of Medicine, Kaohsiung 83301, Taiwan; mingchun@cgmh.org.tw; 5Department of Hematology Oncology, E-Da Cancer Hospital and I-Shou University, College of Medicine, Kaohsiung 82445, Taiwan; sungnanpei@gmail.com; 6Department of Obstetrics and Gynecology, Kaohsiung Chang Gung Memorial Hospital and Chang Gung University College of Medicine, Kaohsiung 83301, Taiwan; n22370@cgmh.org.tw; 7Chung Shan Medical University School of Medicine, Taichung 40201, Taiwan; 8Division of Thoracic and Cardiovascular Surgery, Department of Surgery, Kaohsiung Chang Gung Memorial Hospital and Chang Gung University College of Medicine, Kaohsiung 83301, Taiwan; fanyenlee2015@gmail.com; 9Division of Cardiovascular Surgery, Department of Surgery, Tri-Service General Hospital, National Defense Medical Center, Taipei 11490, Taiwan; 10Department of Nursing, Asia University, Taichung 41354, Taiwan; 11Institute for Translational Research in Biomedicine, Kaohsiung Chang Gung Memorial Hospital and Chang Gung University, College of Medicine, Kaohsiung 83301, Taiwan; 12Department of Medical Research, China Medical University Hospital, China Medical University, Taichung 40402, Taiwan

**Keywords:** diffuse coronary artery disease, angiogenesis, CD34+ cells, preserved LVEF, heart failure, angina, dyspnea

## Abstract

This phase II randomized controlled trial tested whether intracoronary autologous CD34+ cell therapy could further improve left ventricular (LV) systolic function in patients with diffuse coronary artery disease (CAD) with relatively preserved LV ejection fraction (defined as LVEF >40%) unsuitable for coronary intervention. Between December 2013 and November 2017, 60 consecutive patients were randomly allocated into group 1 (CD34+ cells, 3.0 × 10^7^/vessel/*n* = 30) and group 2 (optimal medical therapy; *n* = 30). All patients were followed for one year, and preclinical and clinical parameters were compared between two groups. Three-dimensional echocardiography demonstrated no significant difference in LVEF between groups 1 and 2 (54.9% vs. 51.0%, respectively, *p* = 0.295) at 12 months. However, compared with baseline, 12-month LVEF was significantly increased in group 1 (*p* < 0.001) but not in group 2 (*p* = 0.297). From baseline, there were gradual increases in LVEF in group 1 compared to those in group 2 at 1-month, 3-months, 6-months and 12 months (+1.6%, +2.2%, +2.9% and +4.6% in the group 1 vs. −1.6%, −1.5%, −1.4% and −0.9% in the group 2; all *p* < 0.05). Additionally, one-year angiogenesis (2.8 ± 0.9 vs. 1.3 ± 1.1), angina (0.4 ± 0.8 vs. 1.8 ± 0.9) and HF (0.7 ± 0.8 vs. 1.8 ± 0.6) scores were significantly improved in group 1 compared to those in group 2 (all *p* < 0.001). In conclusion, autologous CD34+ cell therapy gradually and effectively improved LV systolic function in patients with diffuse CAD and preserved LVEF who were non-candidates for coronary intervention (Trial registration: ISRCTN26002902 on the website of ISRCTN registry).

## 1. Introduction

Despite state-of-the-art management strategies for coronary artery disease (CAD) including pharmacomodulation [1,2], continuous education [3], guideline renewal [4], instrument refinement [4,5,6,7], refined technique in percutaneous coronary intervention (PCI) [8,9,10] and the matured operative procedure of coronary artery bypass surgery (CABG) [3,11], it remains the leading cause of death in the last decade. Additionally, the rapidly growing global economic burden for the treatment of CAD [12] further underscores the need for a novel, safe, and effective therapeutic alternative.

Clinical observational studies have revealed that a dominant number of patients with CAD, estimated to be up to 15–20%, were afflicted with severe and diffuse atherosclerotic obstructive CAD who are not only non-candidates for percutaneous or surgical interventions [13,14] but were also poor responders to medical therapy [15]. Additionally, recent study has revealed that incomplete myocardial revascularization closely links to higher ischemic and bleeding risks as compared with a complete revascularization strategy [16]. Moreover, this high-risk patient population without any revascularization has been shown to have the poorest long-term clinical outcomes [17,18].

Growing evidence from clinical trials has demonstrated the safety and effectiveness of autologous CD34+ cell therapy [19,20] for treating ischemia-related left ventricular (LV) dysfunction. Consistently, our phase I clinical trial has also shown that intracoronary (IC) administration of autologous CD34+ cells significantly improved not only short-term [21] but also long-term [22] ischemia-related LV dysfunction as well as symptoms of angina and heart failure (HF). Interestingly, within subgroup analysis, we found no data regarding the therapeutic benefits of endothelial progenitor cells (EPCs) for patients with diffuse CAD and preserved LV ejection fraction (LVEF) who were unsuitable for coronary intervention despite their associated symptoms of angina and dyspnea. This laid the foundation for the current study that investigated whether IC administration of autologous CD34+ cells would improve cardiac function and symptoms of HF and angina in patients with diffuse CAD and relatively preserved LV systolic function unsuitable for coronary intervention.

## 2. Materials and Methods

### 2.1. Study Design

The study design has been clearly described in our recent reports [21,22]. In detail, this was a prospective clinical study performed in a tertiary medical center of southern Taiwan between December 2013 and November 2017. This phase II clinical trial was approved by Taiwan Food and Drug Administration (TFDA) (IRB No: 1066062944), Ministry of Health and Welfare, Taiwan, Republic of China, and the Institutional Review Committee on Human Research at Chang Gung Memorial Hospital (201003985A0) in December 2013 and conducted at Kaohsiung Chang Gung Memorial Hospital, a tertiary referral center. The trial had been registered as ISRCTN26002902 on the website of ISRCTN registry.

This was a prospective, randomized open-label controlled phase II clinical trial to test the safety and efficacy of circulation-derived CD34+ cell treatment for patients with diffuse CAD and relatively preserved LV systolic function in a single tertiary medical center. This study was designed to consecutively enroll 60 patients who had diffuse CAD and intractable angina pectoris with relatively preserved LV function (i.e., defined as LVEF >40%) and were unsuitable for percutaneous catheter-based (PCI) or surgical (CABG) coronary intervention. The patients were randomized to receive either CD34+ cells (3.0 × 10^7^) treatment/per vessel (group 1) or serve as controls with only standard pharmacotherapy (group 2) (i.e., 1:1 randomization) (Figure 1). Additionally, diffuse CAD were defined as more than 20-mm-long stenotic lesions over ≥2 coronary arteries with severely diseased small branches and distal run-off vessels. After heart-team evaluation, those patients unsuitable for coronary intervention or predictably unfavorable clinical outcomes were suggested by investigators to receive the cell-based therapy. The criteria of “non-candidacy” for revascularization included too small luminal diameter of diseased epicardial vessels (defined as <2.5 mm in the proximal segment or <2.0 mm in the distal segment), high risk for myocardial revascularization therapy (e.g., fragile or weak patient and multiple comorbidities), or a patient’s unwillingness. The final decision for cell-based therapy was dependent upon consensus from cardiologist and cardiovascular surgeon as well as discussion with patient and family. 

The primary endpoints of this clinical trial were to test the safety and the improvement in LV function (i.e., efficacy) in group 1 compared with those in group 2. Secondary endpoints included (1) overall survival rate; (2) incidence of CD34+ cell transfusion-related clinical adverse event; (3) significant symptomatic improvement in the degrees of angina pectoris assessed by the Canadian Cardiovascular Society (CCS) Angina Grade Score and heart failure (HF) assessed by the New York Heart Association Functional Classification (NYHA Fc).

### 2.2. Calculation of Rational Sample Size for Endpoints

We calculated sample size of 34 patients in each group on the basis of the effective size with an α = 0.05, a power of 80% and an anticipation of LVEF improvement of 7.0% ± 4.0% in group 1 compared with that in group 2 after assuming the rate of 4.0% for protocol violation and incomplete follow-up. The presumed improvement in LVEF was based on our own [21] and previous [23] publications. 

### 2.3. Inclusion and Exclusion Criteria

The inclusion and exclusion criteria have been described in our previous report [21]. In details, the inclusion criteria included patients (>20 and <80 years old) who had diffuse obstructive CAD (including stable/unstable angina, prior myocardial infarction >3 months or elective stenting for stable CAD) with coronary angiographic findings of at least one severe diffuse CAD, noncandidates for PCI or CABG after heart team approach, those who had CCS Grade II-IV angina, and those with reversible myocardial ischemia shown on thallium (Tl-201) scan. Patients with history of the following conditions were excluded from the study: surgery, trauma, myocardial infarction or stenting within the preceding 3 months, liver cirrhosis, hematology disorders, renal insufficiency (defined as creatinine clearance <20 mL/min), malignancy, febrile disorders, acute or chronic inflammatory disease at study entry, severe mitral or aortic regurgitation, active congestive heart failure (NYHA Fc 4), life expectancy <2.0 years, aged <20 or ≥80 years, or pregnant women. 

An overview of patients’ screening, enrollment, allocation and follow-up is shown in Figure 1. From December 2013 through November 2017, patients who met the criteria were enrolled consecutively at our institute after signing informed consent. Over a 48-month enrollment period, a total of 65 patients with severe diffuse CAD were screened and recruited. Five of the 65 patients (7.7%) were excluded because they refused to participate in the study. A total of 60 patients were equally randomized into group 1 (i.e., CD34+ cells, 3.0 × 10^7^/vessel) and group 2 (i.e., treated with guideline-directed anti-ischemic and anti-HF pharmacotherapy) (Figure 1). 

The relevant investigators, including the echocardiographers, technicians, clinical nurses and physicians taking care of the patients in outpatient clinics, were blinded to the randomization and allocation. Only the cardiologists responsible for coronary intervention as well as the hematologist and technician who were responsible for EPC isolation were unblinded to the study.

### 2.4. Procedure and Protocol for Cell Isolation and Intracoronary Autologous CD34+ Cell Therapy

The validation of the flow cytometric method as well as generation of the final protocols was performed in scope of our previous Phase-I study in 2015 [21]. In detail, before isolation of circulation-derived CD34+ cells, granulocyte-colony stimulating factor (G-CSF) (5 µg/kg, per 12 h for 4 days, a total of 8 doses) was subcutaneously administered to group 1 patients to stimulate the number of circulating CD34+ cells for subsequent cell collection with leukapheresis. After the last dose (i.e., 8th dose) of G-CSF, the number of CD34+ cells in the mononuclear cell preparation isolated during leukapheresis was enriched by utilizing a commercially available device (COBE Spectra 6.1 (Terumo BCT, Inc., Lakewood, CO, USA)) at 8:00 a.m. through a double lumen catheter inserted into the right femoral vein. 

In detail, the COBE Spectra system was set-up and it could maintain the interface by defining the pump flow rates and centrifuge speed based on patient data for the leukapheresis procedure. The COBE Spectra system contains several advantages for CD34+ cell isolation: (1) Separation of components based on specific gravity of cells, (2) Programmable fluid balance and automated procedure endpoint calculations, (3) low total blood volume applications and (4) automated procedures with operator control for adequate numbers of CD34+ cells. After a time-interval about 4 h, adequate circulatory-derived CD34+ cells were collected and well-prepared for intracoronary infusion. 

According to the International Society of Hematotherapy and Grafting Engineering (ISHAGE) Guidelines for CD34+ cell determination with flow cytometric measurement of circulating CD34+ cells, hematological stem cells are characterized by the presence of the surface markers CD34 high/CD45^dim^/SSClow that were used to quantify the number of isolated CD34+ cells. The formula for the number of circulation-derived CD34+ cells was: Number of CD34+ cells = (percentage of CD34+ cells) × white blood cell count × 10^3^ × peripheral-blood stem cell (PBSC) volume (mL). In this phase II trial, flow cytometric analysis followed the current guidelines of the College of American Pathology with a performance coefficient of variation (CV) <4.0% (3.4 ± 2.5) (by definition, CV <10.0% is acceptable).

After CD34+ isolation was completed, the patients were sent to a cardiac catheterization room within 2 h to receive the intra-coronary CD34+ cell transfusion. Trans-radial arterial approach was utilized for each patient in group 1 for coronary angiographic study, followed by slow infusion of target-dose CD34+ cells via a microcatheter into each target vessel of ischemic myocardium. Additionally, right internal jugular vein puncture and implantation of infusion catheter into the coronary sinus (CS) were performed for estimation of the serial changes of EPCs in the venous samplings.

### 2.5. Flow Cytometric Assessment of Circulating and Coronary Sinus EPC Levels and ELISA Evaluation of Soluble Angiogenesis Factors

The procedure and protocol have been described in our previous study [21]. In detail, EPC populations in circulation and coronary sinus were identified by flow cytometry using double staining through fluorescence-activated cells (FC500 Cytometer, Beckman Coulter, Brea, CA, USA). Each analysis included 300,000 cells per sample. The assays for circulatory and CS EPCs in each sample were performed in reproduction, and mean levels were reported. Intra-assay variability was low with a mean CV of 3.9% among the study subjects after repeated measurements of the same blood sample. CXP Analysis software (Beckman Coulter, Brea, CA, USA) was performed for flow cytometry analysis.

One blood sample was extracted at 8:00 a.m. prior to G-CSF injection and the other was collected following final G-CSF treatment for flow cytometric analysis. Additionally, to elucidate the serial changes in the levels of EPC in CS, serial blood samples were drawn from the CS prior to CD34+ cell transfusion and at 5, 10, and 30 min after CD34+ cell transfusion, and then sent for flow cytometric analysis. 

Circulating levels of vascular endothelial growth factor (VEGF), hepatocyte growth factor (HGF), and stromal cell-derived growth factor (SDF)-1α, three indicators of soluble angiogenesis, were measured by duplicated determination with a commercial ELISA method (R&D Systems, Minneapolis, MN, USA). Intra-observer variability of the measurements was also evaluated, and the mean intra-assay CV were all <4.5%. The concentration of serum troponin I (normal range: <0.5 ng/mL) was measured by standard method in the Department of Clinical Biochemistry and Pathology of our hospital.

### 2.6. Coronary Angiographic, Imaging and Laboratory Studies

The procedure and protocol were described as our previous report [21]. We used the standard projective views for the same vessel via fluoroscopic coronary angiography before and 9 months after CD34+ cell therapy for all study subjects in groups 1 and 2. Standardized right anterior-oblique and right cranial or caudal views were utilized for surveying the left coronary system, while the left cranial view was employed for viewing the right coronary system. In addition, a similar volume of contrast (usually <10 mL) was used for each vessel injection with the same cine angiographic time. 

Nine months later, we performed the follow-up angiography for the coronary arterial trees with the standard four projective views and used the Wimasis Image Analysis System (Wimasis GmbH: Limited Liability Company, Munich, Germany) for the analysis of neovascularization. 

The 2-D and 3-D transthoracic echocardiography were performed at 3, 6 and 12 months by an experienced cardiologist blinded to the patient grouping. The procedure and protocol of 3-D transthoracic echocardiography were previously reported [24].

### 2.7. Definition for Angiographic Angiogenesis Score

The definition has been described in our previous reports [21,22]. In detail, the semi-quantitative angiographic grading of angiogenesis/neovascularization (i.e., angiographic score of vessel density pre-CD34+ and 9-month post-CD34+ cell therapy) was defined as: Grade 0: <5%; Grade 1: 5–35%; Grade 2: >35–75%; Grade 3: >75%. Additionally, we utilized a scientific method of Wimasis Image Analysis (Onimagin Technologies SCA, Córdoba, Spain) for quantitative analysis of angiogenesis.

### 2.8. Medications

Heparin (3000 IU) was intra-arterially given to each patient at the beginning of the procedure and its effect was immediately reversed by intra-venous administration of 15 mg of protamine after CD34+ cell transfusion. The purpose of protamine infusion was to prevent bleeding complication in the arterial and venous accesses and to mitigate the heparin-related alternation of stem cell responsiveness. Aspirin was prescribed for all patients unless they were allergic or intolerant to aspirin owing to gastric ulcer, duodenal ulcer or upper gastrointestinal tract bleeding, or hyperreactivity to aspirin. If so, clopidogrel or ticagrelor was alternatively used in those patients with intolerance to aspirin therapy. Other common medications included statin, beta blocker, angiotensin converting enzyme inhibitor (ACEI)/angiotensin II type I receptor blocker (ARB), diuretic, calcium channel blocker and isosorbide dinitrate that were administered as guideline recommendation.

### 2.9. One-Year Follow-up for Clinical Outcomes

During regular follow-up of each patient at our outpatient clinic, a case report form that recorded all patients’ clinical information, including the presence or absence of acute or subacute adverse events, was used for each study subject and regularly fulfilled by a research nurse after each visit or on readmission, as well as through telephone interviews on an irregular basis.

### 2.10. Statistical Analysis

Final results were analyzed with an intention-to-treat protocol. All variables are expressed as mean ± standard deviation or number with percentage as appropriate. Paired t test was utilized in the same group for determining the significance of changes in continuous variables at different time points. Significance of fluctuations in continuous variables over different time points within the same group was evaluated using repeated measures analysis of variation (RMANOVA). Additionally, independent t test was performed for comparison of parametric continuous variables between the two groups, including baseline data and outcome assessment. In contrast, those continuous variables without normal distribution were compared with Mann–Whitney U test. Statistical analysis was performed using SPSS statistical software for Windows version 19 (SPSS for Windows, version 19; SPSS, IL, USA). A *p* value <0.05 was considered statistically significant.

## 3. Results

### 3.1. Baseline Characteristics of Group 1 and Group 2 Patients

As shown in Table 1, the age, body weight, body mass index and prevalence of hypertension, diabetes mellitus, old stroke and myocardial infarction did not differ between group 1 (i.e., study group) and group 2 (i.e., control group) patients. However, the body height as well as the prevalence of male gender and hyperlipidemia were significantly higher in group 1 than those in group 2.

The rates of left main disease and triple vessel disease as well as history of CABG were similar between the two groups. However, the histories of in-stent restenosis with a need for stenting and catheter-based coronary intervention were significantly higher in group 1 than those in group 2. 

Laboratory examinations demonstrated no significant differences in white blood cell count, platelet count, hemoglobin, creatinine clearance rate, liver function, total cholesterol, low- and high-lipoprotein, hemoglobin, serum creatinine and triglyceride levels between the two groups. 

Furthermore, there was no difference in the use of lipid-lowering agents, angiotensin II type I receptor blockers/angiotensin converting enzyme inhibitor, β-blocker agents, diuresis, calcium channel blocker agents, vasodilator and anti-platelet agents between the two groups. Besides, the two groups expressed a similar rate of endothelial dysfunction measured by flow-mediated dilatation (FMD) of the brachial artery (defined as post-nitroglycerin FMD <300%).

### 3.2. Clinical and Angiographic Findings and Prognostic Outcomes

The Table 2 demonstrates that the mean number of vessels treated by CD34+ cell infusion was 1.72 ± 0.53, including 31% one-vessel, 65.5% two-vessel and 3.4% three-vessel treatments. Troponin-I level was found to be a little bit higher after CD34+ cell administration than the normal standard (i.e., <0.3 ng/mL) in healthy subjects.

The angina score and HF functional class did not differ between the two groups prior to CD34+ cell therapy. However, the two parameters were significantly lower in group 1 than those in group 2 at 1, 3, 6 and 12 months after CD34+ cell treatment. Additionally, as compared with the baseline, these parameters were significantly decreased in group 1 than those in group 2 at 12-month following cell infusion. Furthermore, angiogenesis score on coronary angiographic study at 9-month follow-up was significantly higher in group 1 than that in group 2. 

The incidences of unfavorable clinical outcomes, including all-cause mortality, major adverse cardiovascular or cerebrovascular event (MACCE) and hospitalization for HF, did not differ between the two groups. Looking closer, all-cause mortalities in four patients of group 1 were traumatic brainstem hemorrhage, pneumonia and suffocation with hypoxic respiratory failure, sepsis and septic shock, and sudden cardiac death, respectively. On the other hand, only one patient in group 2 expired at a local hospital due to dengue hemorrhagic fever with septic shock and multiple organ failure. 

More patients in group 2 than those in group 1 received salvage catheter-based coronary revascularization for intractable angina during follow-ups (i.e., mostly at 6 or 9 months after enrollment) in an attempt to relieve ischemic symptoms by deploying one or two stents at the severely stenotic lesion(s). However, the difference in incidence of this interventional procedure between the two groups did not reach statistical significance.

### 3.3. Comparison of Circulating EPC Surface Markers and Soluble Angiogenesis Factors Between Groups 1 and 2 Before and After G-CSF Treatment in Group 1 and Changes in EPC Population and SDF-1 α Concentration in Coronary Sinus (CS) in Group 1 at Different Time Points

Figure 2 reveals that flow cytometric analysis was utilized to confirm the surface markers of EPC post G-CSF stimulation. The circulating populations of EPC (i.e., CD34+ KDR+ CD45^dim^, CD34+ CD133+ CD45^dim^, CD31+ CD133+ CD45^dim^, CD34+ CD133+ KDR+, CD133+) and hematopoietic stem cell (HSC) (CD34+) did not show significant difference between groups prior to G-CSF treatment but were significantly increased in group 1 after administration of G-CSF or in plasma containing isolated EPCs (Figure 2A–F). 

In addition, the results of ELISA showed that the circulating levels of VEGF and HGF, two soluble pro-angiogenic factors, were significantly increased after G-CSF treatment in group 1 patients (Figure 2G,H).

Intriguingly, ELISA demonstrated that the level of SDF-1α in coronary sinus was increased significantly post G-CSF treatment, and further elevated in plasma-containing isolated EPCs compared to that prior to G-CSF treatment (Figure 2I).

Flow cytometric analyses demonstrated continuous drainage of EPCs from CS to circulation after intra-coronary administration (Figure 2J).

### 3.4. Objective Evaluation of Angiogenesis with Wimasis Software

For a more objective assessment, we utilized the Wimasis software (Onimagin Technologies SCA, Córdoba, Spain, https://www.wimasis.com/en/) to analyze the parameters of angiogenesis, including vessel density, total vessel network length, total branching points, total nets, total segments and segment length (refer to Table 3 and Figure 3). At baseline, these parameters did not differ between group 1 and group 2 patients. However, by the ninth month after cell therapy, total vessel network length, total branching points and total segments were significantly increased in group 1 compared to those in group 2.

### 3.5. Changes in LVEF Compared with Baseline and Serial Changes on 3-D Echocardiography During One-Year Follow up

The serial changes in LVEF on 3-D echocardiography were schematically illustrated in Figure 4A–C. The results demonstrated a stepwise increase in LVEF during follow-ups at different time points (i.e., 0, 1, 3, 6, and 12 months) in group 1 (Figure 4A). In contrast, LVEF decreased initially and reached a plateau in group 2 (Figure 4A). 

The results of 3-D echocardiography showed no difference in LVEF between the two groups at any time point (Figure 4B). However, as compared with baseline, this parameter was significantly increased among group 1 patients by the sixth month and the twelfth month after cell therapy, but it did not differ among group 2 patients at these time points (Figure 4B). 

Additionally, net improvements in LVEF between baseline and 1 month (1.58 ± 4.88 vs. −1.56 ± 3.13, *p* = 0.023), 3 months (2.16 ± 8.77 vs. −1.50 ± 2.16, *p* = 0.035), 6 months (2.86 ± 2.26 vs. 1.38 ± 3.49, *p* < 0.001) and 12 months (4.63 ± 2.51 vs. −0.94 ± 4.57, *p* < 0.001) were significantly more pronounced in group 1 than those in group 2 (Figure 4C).

### 3.6. Matrigel Assay for Assessment of Angiogenesis

Figure 5 shows that the results of Matrigel assay demonstrated that at baseline the parameters of angiogenesis, including number of tubular formation, total tubular length/mean tubular length, number of cluster formation and number of network formation, did not differ between the two groups. However, these parameters were significantly increased in G-CSF-treated group 1 patients and further significantly increased in plasma containing isolated CD34+ cells in group 1 patients than those in groups 1 and 2 patients at baseline. The above findings implied that CD34+ cell therapy improved clinical presentation of dyspnea or angina, LVEF, and angiographic angiogenesis score through an increase in angiogenesis. 

### 3.7. Comparison of Echocardiographic Parameters Between Two Groups at Baseline and 12 Months (Appendix A)

The Appendix A lists the 2-D and 3-D echocardiographic parameters of groups 1 and 2 patients. There was no significant difference between the two groups.

### 3.8. Illustrating the Example of Flow Cytometric Analysis how to Gate the EPC Surface Markers (Appendix A)

### 3.9. Illustrating the Presentation of the Backing Gate for EPC Surface Maker (Appendix A)

## 4. Discussion

There were several clinically important implications in the present study. Frist, the novel finding in this phase II clinical trial was that IC CD34+ cell therapy still offered an additional improvement in LVEF in patients with relatively preserved cardiac performance even in the presence of diffuse CAD. Second, the net improvements in LVEF at different time points were found to occur only in those after CD34+ cell treatment. The symptoms of angina and dyspnea were significantly improved in group 1 patients treated with CD34+ cells but not in group 2 patients treated with standard medications at time points of 1, 3, 6 and 12 months.

Interestingly, while growing data support the benefit of EPC infusion in terms of improving LVEF in patients with ischemia-related LV systolic dysfunction [19,20,23], its therapeutic impact on LVEF in patients with relatively preserved cardiac performance and diffuse CAD was unclear. The unique finding in the present study was that LVEF was notably improved in group 1 patients who had relatively preserved baseline LV function following CD34+ cell treatment. Accordingly, our finding encourages the use of EPCs for those with LVEF >40% and angina/dyspnea symptoms but refractory to optimal medical treatment.

A number of previous studies have clearly demonstrated that EPC therapy remarkably improved angina and HF symptoms in patients with diffuse CAD unsuitable for coronary intervention [19,20,21,22,23]. An essential finding in the present study was that, as compared with group 2 patients, the degrees of angina and HF severity were significantly and continuously improved in group 1 patients after CD34+ cell therapy. In this way, our findings were consistent with those of previous studies [19,20,21,22,23]. 

Our phase I clinical trial demonstrated that the 1-year and 5-year angiogenesis scores in patients with severe diffuse CAD were significantly improved after CD34+ cell therapy [21,22]. Another key finding in the current study was that angiogenesis was significantly increased in group 1 compared to that in group 2 from coronary angiographic assessment and Wimasis software analysis. Accordingly, our findings, in addition to being consistent with those of our previous studies [21,22], could at least in part explain the remarkable improvements in clinical and echocardiographic parameters (i.e., angina, HF, and LVEF) in group 1 compared to those in group 2.

Additionally, the circulating levels of VEGF and HGF (i.e., two soluble angiogenesis factors) were also found to be augmented in group 1 patients after G-CSF treatment. One distinctive finding was that SDF-1α concentration (i.e., a kind of chemokine for EPCs mobilization from bone marrow to circulation and homing to ischemic region) was remarkably higher in CS and furthermore increased in plasma containing isolated CD34+ cells than that in circulation, suggesting that a higher concentration of SDF-1α in coronary arteries could trap EPCs within the coronary artery trees/micro-vasculatures (i.e., an effect of ligand-receptor binding for angiogenesis and neovascularization). Furthermore, the circulating EPC population (Figure 2) and Matrigel assay of angiogenesis parameters (Figure 3 and Table 3) were significantly increased after G-CSF treatment in group 1 patients. These findings, once again, explained the improvements in heart function and symptoms of angina after CD34+ cell treatment in group 1 patients. 

In light of current positive findings on the improvement of patients’ heart function and clinical presentation, cell-based therapy can be considered as an effective and safe alternative therapy for patients with severe diffuse CAD and preserved LV function. Together with the findings from our previous researches [21,22], intracoronary CD34+ cell therapy plus standard medical treatment could be suggested to this kind of high-risk patient who has no choice beyond percutaneous or surgical intervention for their severe diffuse coronary lesions. This study has limitations. First, the relatively small sample size could distort statistical significance of some parameters. Second, double-blinded randomization of G-CSF treatment and catheter-based injection of pure plasma in control patients were not permitted by TFDA. Therefore, therapeutic bias could not be completely excluded. Third, current anti-HF medications potential for diastolic dysfunction, e.g., ARB, spironolactone and sodium-glucose transport protein 2 inhibitor, were not investigated for their impact on the LVEF improvement. Finally, due to inadequate financial support, long-term follow-up and collection of additional parameters were not feasible. 

## 5. Conclusions

The results of the present study demonstrated that IC administration of CD34+ cells could further improve left ventricular systolic function even in patients with severe diffuse CAD and relatively preserved cardiac function.

## Figures and Tables

**Figure 1 jcm-09-01043-f001:**
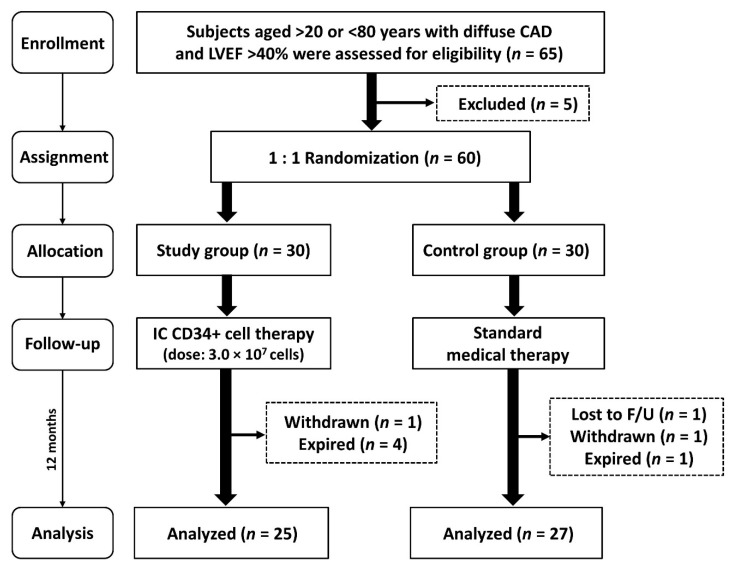
The flow chart showing the enrollment, assignment, allocation, follow-up and analysis in this phase II clinical trial. CAD = coronary artery disease; IC = intracoronary artery; LVEF = left ventricular ejection fraction; F/U = follow.

**Figure 2 jcm-09-01043-f002:**
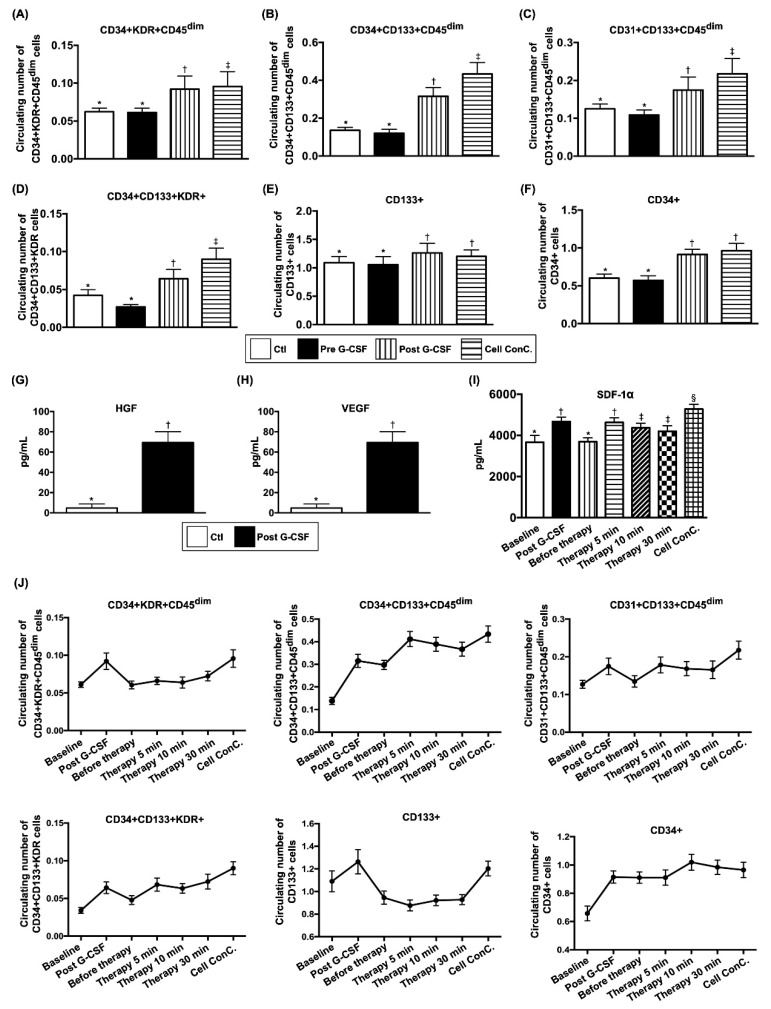
Circulating EPC surface markers and soluble angiogenesis factors between group 1 and group 2 patients prior to and in group 1 after G-CSF treatment, and the time courses of EPC populations and SDF-1 α concentration in coronary sinus (CS) among group 1 patients. (**A**) Circulating number of CD34+ KDR+ CD45^dim^ cells, * vs. other groups with different symbols (†, ‡), *p* < 0.0001. (**B**) Circulating number of CD34+CD133+CD45^dim^ cells, * vs. other groups with different symbols (†, ‡), *p* < 0.0001. (**C**) Circulating number of CD31+ CD133+ CD45^dim^ cells, * vs. other groups with different symbols (†, ‡), *p* < 0.0001. (**D**) Circulating number of CD34+ CD133+ KDR+ cells, * vs. other groups with different symbols (†, ‡), *p* < 0.0001. (**E**) Circulating number of CD133+ cells, * vs. †, *p* < 0.001. (**F**) Circulating number of CD34+ cells, * vs. †, *p* < 0.0001. Clt = control group (i.e., group 1); pre-G-CSF = indicated prior to G-CSF treatment in study group (SG) (i.e., group 1). (**G**) ELISA result of circulating hepatocyte growth factor (HGF), * vs. †, *p* < 0.0001. (**H**) ELISA result of circulating vascular endothelial growth factor (VEGF), * vs. †, *p* < 0.0001. SG (study group) (i.e., group 1), pre-G-CSF = indicated prior to G-CSF treatment in SG; post = indicated post G-CSF treatment in SG. (**I**) The baseline and the time courses of stromal cell-derived factor (SDF)-1α in CS, * vs. other groups with different symbols (†, ‡, §), *p* < 0.001. (**J**) The baseline and the time courses of EPCs populations in coronary sinus, *p* for trend <0.001 for each EPC surface marker. All statistical analyses were performed by one-way ANOVA, followed by Bonferroni multiple comparison post hoc test (*n* = 30 for each group). Symbols (*, †, ‡) indicate significance (at 0.05 level). G-CSF = granulocyte-colony stimulating factor.

**Figure 3 jcm-09-01043-f003:**
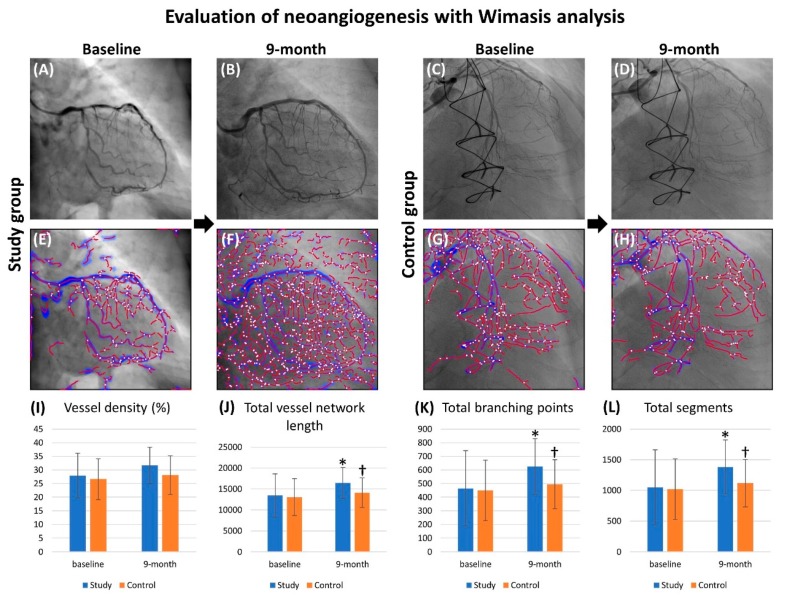
Illustrating the angiographic findings for identification of angiogenesis and Wimasis image analysis for assessment of angiogenesis. (**A**,**D**) Illustrating the coronary angiographic findings prior to (**A**), (**C**) and at 9th month (**B**), (**D**) after CD34+ therapy. As compared to control group (**D**), abundant angiogenesis/neovascularization were notably increased in study group (**B**) at 9th month after CD34+ therapy in both the same patients. (**E**,**H**) Illustrating regional microvasculature angiogenesis level in fixed territory assessment with Wimasis image software, between these two-time intervals in study group and control group. As compared to control group (**H**), plentiful angiogenesis/neovascularization (red color) were noted as having increased in study group (**F**) at 9th month after CD34+ therapy in both the same patients. (**I**) Wimasis assay for angiogenesis parameter of vessel density: (1) at baseline and (2) at 9th month, all *p* > 0.5. (**J**) Wimasis assay for angiogenesis parameter of total vessel network length: (1) at baseline, *p* > 0.5; (2) at 9th month, * vs. †, *p* < 0.05. (**K**) Wimasis assay for angiogenesis parameter of total branch point: (1) at baseline, *p* > 0.5; (2) at 9th month, * vs. †, *p* < 0.05. (**L**) Wimasis assay for angiogenesis parameter for total segments: (1) at baseline, *p* > 0.5; (2) at 9th month, * vs. †, *p* < 0.05.

**Figure 4 jcm-09-01043-f004:**
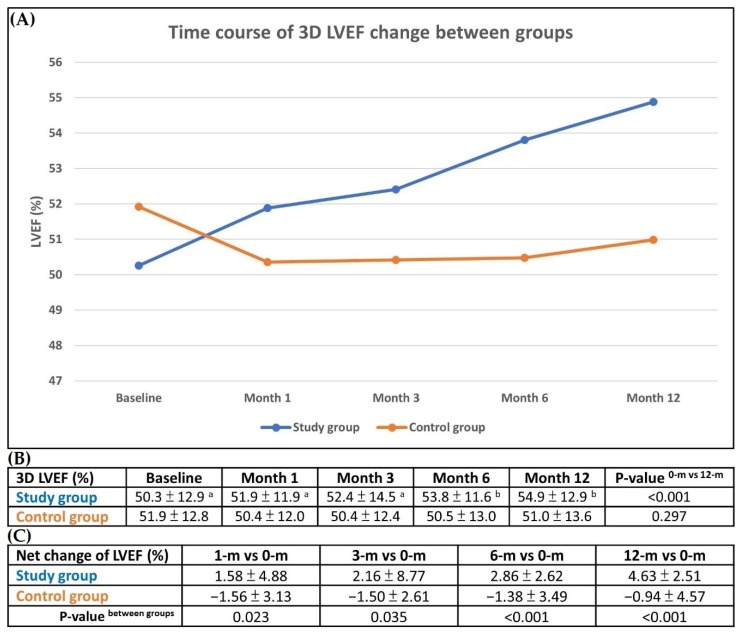
Individual and mean changes of LVEF between baseline and 12 months, and serial changes of LVEF on 3-D echocardiographic findings during 1-year follow-up. (**A**) Illustrating the time courses of 3-D transthoracic echocardiographic findings of left ventricular ejection fraction (LVEF) among group 1 and group 2 patients. The results demonstrated that the LVEF was notably stepwise increased during the follow-up time points at 0, 1, 3, 6 and 12 months in group 1 patients. However, this parameter showed initially downwards, followed by stationary no more change in group 2 patients. (**B**) Compared with baseline (i.e., 0 month), the LVEF was significantly increased among group 1 patients, *p* < 0.001, but it showed no difference among group 2 patients, *p* = 0.29. (**C**) Illustrating the net LVEF improvements at different time points were significantly higher in group 1 than in group 2. Increases in the net change of LVEF improvement in group 1 as compared to group 2 at the time intervals between baseline vs. 1 month: +1.58 ± 4.88 vs. −1.56 ± 3.13, *p* = 0.023; 3 month: +2.16 ± 8.77 vs. −1.50 ± 2.16, *p* = 0.035; 6 month: +2.86 ± 2.26 vs. −1.38 ± 3.49, *p* < 0.001; 12 month: +4.63 ± 2.51 vs. −0.94 ± 4.57, *p* < 0.001.

**Figure 5 jcm-09-01043-f005:**
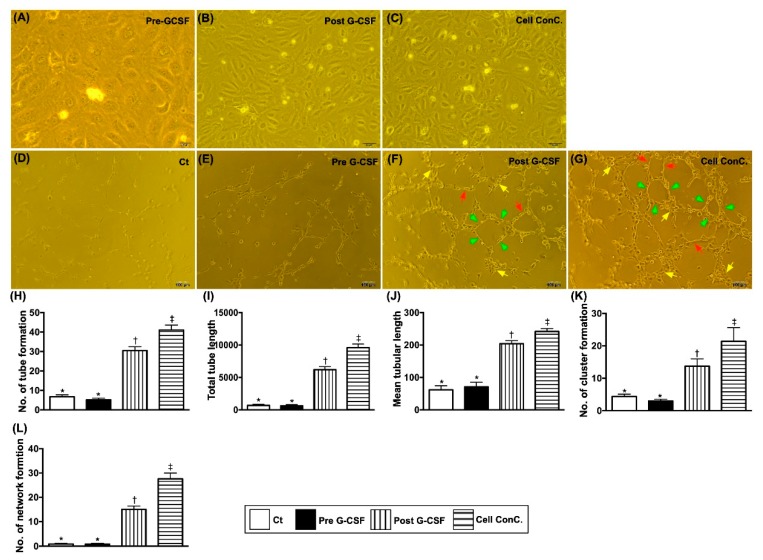
Morphological feature of EPCs and Matrigel assay for assessment of angiogenesis. (**A**–**C**) Illustrating the microscopic finding (200x) of morphological feature of 21-day culturing endothelial progenitor cells (EPCs), i.e., cobblestone-like morphology of typical endothelial cells. Scale bars in the right lower corner represent 50 µm. (**D**–**G**) Illustrating the pictures of Matrigel assay. Red arrows indicated tubular formation; yellow arrows indicated cluster formation; green arrows indicated network formation. (**H**) Analytical result for number of tubule formation, * vs. other groups with different symbols (†, ‡), *p* < 0.0001. (**I**) Analytical result of total tubular length, * vs. other groups with different symbols (†, ‡), *p* < 0.0001. (**J**) Analytical result of mean tubular length, * vs. other groups with different symbols (†, ‡), *p* < 0.0001. (**K**) Analytical result of number of cluster formation, * vs. other groups with different symbols (†, ‡), *p* < 0.0001. (**L**) Analytical result of number of network formation, * vs. other groups with different symbols (†, ‡), *p* < 0.0001. All statistical analyses were performed by one-way ANOVA, followed by Bonferroni multiple comparison post hoc test (*n* = 30 for each group). Symbols (*, †, ‡) indicate significance (at 0.05 level). Clt = control group; pre-G-CSF = the study group prior to G-CSF treatment; post G-CSF = indicated the study group of post G-CSF treatment; G-CSF = granulocyte-colony stimulating factor; No. = number.

**Table 1 jcm-09-01043-t001:** Baseline Characteristics.

Variables	Study Group (*n* = 30)	Control Group (*n* = 30)	*p*-Value
Clinical information			
Age, year	64.57 ± 8.00	65.77 ± 7.29	0.546
Male sex, *n* (%)	28 (93.3%)	22 (73.3%)	0.038
Body height, cm	163.87 ± 12.53	160.37 ± 7.48	0.036
Body weight, kg	70.71 ± 10.71	72.65 ± 15.56	0.576
Body mass index, kg/m^2^	26.58 ± 5.11	28.18 ± 5.47	0.114
Smoker, *n* (%)	11 (36.7%)	12 (40.0%)	0.791
Hypertension, *n* (%)	28 (93.3%)	28 (93.3%)	1.000
Diabetes mellitus, *n* (%)	22 (73.3%)	19 (63.3%)	0.405
Dyslipidemia, *n* (%)	28 (93.3%)	21 (70.0%)	0.020
Old stroke, *n* (%)	8 (26.7%)	6 (20.0%)	0.542
Old myocardial infarction, *n* (%)	5 (16.7%)	5 (16.7%)	1.000
LM involvement, *n* (%)	14 (46.7%)	12 (40.0%)	0.602
Triple vessel CAD, *n* (%)	29 (96.7%)	28 (93.3%)	1.000
In-stent restenosis, *n* (%)	26 (86.7%)	17 (56.7%)	0.010
History of CABG, *n* (%)	10 (33.3%)	13 (43.3%)	0.426
History of PCI, *n* (%)	28 (93.3%)	22 (73.3%)	0.038
Laboratory data			
Leukocyte, 1000/μL	7.31 ± 2.42	6.80 ± 1.83	0.549
Hemoglobin, g/dL	13.73 ± 1.75	13.51 ± 1.82	0.624
Platelet, 1000/μL	210.27 ± 60.08	203.80 ± 55.81	0.673
Serum creatinine, mg/dL	1.25 ± 0.49	1.06 ± 0.30	0.178
eGFR, mL/min	65.18 ± 20.85	70.80 ± 21.36	0.451
Alanine aminotransferase, U/L	22.67 ± 14.17	25.41 ± 13.08	0.255
Total cholesterol, mg/dL	156.50 ± 41.41	151.72 ± 30.35	0.616
Low density lipoprotein	88.90 ± 35.77	81.69 ± 27.61	0.399
High density lipoprotein	42.37 ± 8.85	43.28 ± 8.06	0.682
Triglyceride	137.50 ± 84.27	134.86 ± 73.52	0.891
Endothelial dysfunction *, *n* (%)	17 (56.7%)	16 (53.3%)	0.795
Medications			
Antiplatelet, *n* (%)	30 (100.0%)	30 (100.0%)	1.000
Beta blocker, *n* (%)	28 (93.3%)	28 (93.3%)	1.000
RAAS blocker, *n* (%)	27 (90.0%)	26 (86.7%)	1.000
Calcium channel blocker, *n* (%)	13 (43.3%)	12 (40.0%)	0.793
Diuretic, *n* (%)	9 (30.0%)	8 (26.7%)	0.774
Lipid lowering agent, *n* (%)	22 (73.3%)	23 (76.7%)	0.776
Vasodilator, *n* (%)	17 (56.7%)	22 (73.3%)	0.176

Data are expressed as mean ± standard deviation (SD) or number (percentage). * The endothelial dysfunction was measured with method of flow-mediated dilatation (FMD). Abbreviations: LM = left main; CAD = coronary artery disease; CABG = coronary artery bypass graft; PCI = percutaneous coronary intervention; eGFR = estimated glomerular filtration rate; RAAS = renin-angiotensin-aldosterone system.

**Table 2 jcm-09-01043-t002:** Clinical and Angiographic Findings and Prognostic Outcomes.

Variables	Study Group (*n* = 30)	Control Group (*n* = 30)	*p*-Value
No. of vessel treated by CD34+ cells	1.72 ± 0.53		
1 vessel, *n* (%)	9 (31.0%)		
2 vessels, *n* (%)	19 (65.5%)		
3 vessels, *n* (%)	1 (3.40%)		
Troponin-I after CD34+cell therapy	1.37 ± 4.09		
Scores of angina and HF			
CCS angina score at baseline	2.81 ± 0.54	2.52 ± 0.75	0.325
CCS angina score at 1 months	1.33 ± 0.88	2.53 ± 0.57	<0.001
CCS angina score at 3 months	0.78 ± 0.79	2.25 ± 0.66	<0.001
CCS angina score at 6 months	0.56 ± 0.80	2.26 ± 0.94	<0.001
CCS angina score at 12 months	0.44 ± 0.75	1.81 ± 0.88	<0.001
*p*-value ^12 M vs. 0 M^	<0.001	0.009	
NYHA Fc at baseline	2.07 ± 0.87	1.93 ± 0.83	0.189
NYHA Fc at 1 months	1.32 ± 0.82	1.97 ± 0.67	0.002
NYHA Fc at 3 months	1.00 ± 0.77	2.00 ± 0.62	<0.001
NYHA Fc at 6 months	0.59 ± 0.75	2.07 ± 0.62	<0.001
NYHA Fc at 12 months	0.67 ± 0.83	1.78 ± 0.64	<0.001
*p*-value ^12 M vs. 0 M^	< 0.001	0.377	
Angiogenesis score on 9-month by coronary angiographic study	2.83 ± 0.87	1.32 ± 1.10	<0.001
Clinical outcomes at 1 year			
All-cause mortality, *n* (%)	4 (13.8%)	1 (3.4%)	0.352
MACCE, *n* (%)	3 (10.3%)	3 (10.3%)	1.000
Cardiovascular death	1 (3.4%)	0 (0.0%)	1.000
Acute myocardial infarction	0 (0.0%)	1 (3.4%)	1.000
Acute stroke	2 (6.9%)	2 (6.9%)	1.000
Hospitalization for HF, *n* (%)	3 (10.3%)	0 (0.0%)	0.237
Revascularization, *n* (%)	4 (13.8%)	7 (24.1%)	0.315
Sepsis, *n* (%)	3 (10.3%)	1 (3.4%)	0.611

Abbreviations: CV = Coefficient of Variation; CCS = Canadian cardiovascular society; NYHA Fc = New York Heart Association functional class; CMR = Cardiovascular magnetic resonance imaging; LVEF = left ventricular ejection fraction; CAG = coronary angiography; MACCE = major adverse cardiovascular or cerebrovascular event; HF = heart failure. Data are expressed as mean ± standard deviation (SD) or number (%). Composite endpoint for HF was defined as those suffering all-cause mortality or hospitalization for acute decompensated heart failure.

**Table 3 jcm-09-01043-t003:** Objective evaluation of angiogenesis with Wimasis software analysis.

Variables	Study Group (*n* = 25)	Control Group (*n* = 27)	*p*-Value
Baseline CAG			
Global metrics			
Vessel density, %	27.92 ± 8.20	26.63 ± 7.54	0.530
Total vessel network length, pixel	13438 ± 5200	13064 ± 4377	0.765
Total branching points	465.6 ± 276.9	450.4 ± 221.2	0.815
Total nets	45.77 ± 22.79	46.46 ± 21.32	0.796
Segment characteristics			
Total segments	1049.8 ± 612.9	1019.5 ± 494.8	0.834
Segment length, pixel	15.80 ± 5.42	15.80 ± 4.34	1.000
Follow-up CAG at 9 months			
Global metrics			
Vessel density, %	31.66 ± 6.69	28.13 ± 7.13	0.116
Total vessel network length, pixel	16466 ± 3720	14104 ± 3523	0.033
Total branching points	625.4 ± 204.4	495.2 ± 180.2	0.027
Total nets	49.68 ± 13.51	49.29 ± 17.15	0.524
Segment characteristics			
Total segments	1383.5 ± 439.8	1117.7 ± 385.9	0.035
Segment length, pixel	13.38 ± 1.67	12.71 ± 1.79	0.240

Data are expressed as mean ± standard deviation. Abbreviation: CAG = coronary angiography.

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
