# Peer review of "Intracoronary Injection of Autologous CD34+ Cells Improves One-Year Left Ventricular Systolic Function in Patients with Diffuse Coronary Artery Disease and Preserved Cardiac Performance—A Randomized, Open-Label, Controlled Phase II Clinical Trial"

_jcm, 2020, doi:10.3390/jcm9041043_

Round 1
Reviewer 1 Report
Interesting RCT
Some minor issues
Introduction: recently uncomplete revascualrizartion (PMID: 31809991) has been linked to poor prognosis. This should be quoted.
1) diffuse CAD should be better defined
2) definition of unsuitable for PCI/CABG should be added
3) for sample size it should be added if 7% and 4% derived from some already published reference
4) normal distribution should be evalauted
Author Response
Comments and Suggestions for Authors:
Interesting RCT
Comment 1: Introduction: recently uncomplete revascularization (PMID: 31809991) has been linked to poor prognosis. This should be quoted.
Response 1: Dear reviewer, we have cited the reference [16] as your suggestion. Please refer to the lines 66-68 in the introduction of our revised manuscript. Thank you.
Comment 2:
1) diffuse CAD should be better defined
2) definition of unsuitable for PCI/CABG should be added
3) for sample size it should be added if 7% and 4% derived from some already published reference
4) normal distribution should be evaluated
Response 2: Thank you for the instructions. 1) & 2) We have added the definition of diffuse CAD and unsuitable lesions for PCI/CABG in the lines 99-102 in the method section. 3) The published references have been added in the lines 113-114.
4) All data in the study was assessed and analyzed by an experienced statistician. Examination of normal distribution had been evaluated before data analysis. For easy understanding by readers, we expressed all data with mean ± SD. However, P-values were expressed as appropriate after analysis with independent t test for parametric data and Mann-Whitney U test for nonparametric data. We have added detailed statistical description in the lines 236-238 in the method section. Thank you.
We would like to take this opportunity to express our appreciation for your detailed review of the article and the kindness of giving us valuable suggestions. Thank you very, very much!
Reviewer 2 Report
In the manuscript entitled: “ Intracoronary Injection of Autologous CD34+ Cells Improves One-Year Left Ventricular Systolic Function in Patients with Diffuse Coronary Artery Disease and Preserved Cardiac Performance - A Randomized, Open-label, Controlled Phase II Clinical Trial” Sung and coworkers suggest that autologous CD34+ cell therapy gradually and effectively improved LV systolic function in patients with diffuse CAD and relatively preserved LVEF.
This manuscript addresses the use of autologous CD34+ therapy when the coronary interventions such as PCI or CABG cannot be applied or when poor response to standard medications occurs. Mainly, the issue depicted seems to be relevant towards regenerative medicine, predominantly for patients with a HF-mrEF or even for HF –pEF with no chance to obtain an evidence-based medicine.
However, some minor/major suggestions and additions might be relevant and are required before considering the present manuscript as suitable for publication. It is recommended to make corrections due to some errors/ typing errors and in mentioning the references by numbers which surely were illogical with the sentence in the manuscript at some point.
- In the present study the patient cohort is composed of the patients with HF-mrEF and even with HF-pEF, which is very interesting, since no evident-based medicine is available yet. Overall, only to some extent the applied medications for patients with HF and EF >40% (HF-rEF) revealed benefits. Do the authors have any observations or suggestions to the current application of the SGLT2 inhibitors e.g. Dapagliglozin?
- The gradually improvement of the symptoms after the therapy in comparison to medications seem to be very interesting and provide a new data set. However, it would be yet interesting to show data and alterations with long term course (here >12 months)
- Moreover, it is questioned, how a renunciation of the medicinal therapy, e.g. ASS and statins, may influence on the long term course the risk of a myocardial infarction. For this purpose, collection of the data from the long term therapy with larger number of the patient cohort would be required. In this regard, the cross-regional studies would be advantageous.
- The numbers of the dose do not match over the whole manuscript. Please see the Figure 1.
- Please describe the isolation of CD34+ cells more in details (the line 143), the EPC flow cytometric analysis/ strategy using representative dot plots. Why final 300.000 cells were set up as a threshold for the final analysis and which gate has been chosen?
- Please comment on the use of the protamine and discuss about the current application after CD34+ transfusion.
- How the absolute count of the EPCs ( Figure 2) was calculated? Please describe in detail the entire analysis relying on some representative plots.
- What is the role of these separately analyzed EPCs, please comment on that more detailed.
Author Response
Dear Reviewer:
Thank you so much for your detailed review of our manuscript. Your constructive criticism is greatly appreciated. We have made the following point-by-point responses to comply with your honorable suggestions (Note: The revised parts of the manuscript in response to reviewer’s comments have been marked in blue color).
Comment 1: In the present study the patient cohort is composed of the patients with HF-mrEF and even with HF-pEF, which is very interesting, since no evident-based medicine is available yet. Overall, only to some extent the applied medications for patients with HF and EF >40% (HF-rEF) revealed benefits. Do the authors have any observations or suggestions to the current application of the SGLT2 inhibitors e.g. Dapagliflozin?
Response 1: Dear reviewer, thank you for your expertise and interesting question. We totally agreed with you that current guideline-directed medical anti-HF therapies are mostly beneficial to patients with HFrEF. Only ARB and spironolactone have been shown partially beneficial to HFpEF because the etiology of HFpEF/HFmrEF is diverse and complex. Although the prevalence of diabetes in both groups about 70%, majority of they received DPP4 inhibitors rather than SGLT2 inhibitors. That is because our study period was between 2013 and 2017 in which SGLT2 inhibitor were not yet available in Taiwan. Therefore, we cannot conclude whether SGLT2 inhibitor plays a useful role in the improvement of LVEF for the patients with severe diffuse CAD with HFmrEF or HFpEF. We have described above statement in the lines 389-391 in the limitation section of our revised manuscript. Thank you for your understanding.
Comment 2: The gradually improvement of the symptoms after the therapy in comparison to medications seem to be very interesting and provide a new data set. However, it would be yet interesting to show data and alterations with long term course (here >12 months)
Response 2: Thank you for your informative suggestion. We are also very interested in this analysis because our previous 5-year data from CD34+ cell therapy for severe diffuse CAD with HFrEF (please refer to reference 22) showed stationary change without deteriorating LV function beyond one year. For the current study, the patient’s LVEF beyond 1 year is still followed up by our PI-initiated research owing to limited funding/economic support, and the data will be published to address your question in the future.
Comment 3: Moreover, it is questioned, how a renunciation of the medicinal therapy, e.g. ASA and statins, may influence on the long term course the risk of a myocardial infarction. For this purpose, collection of the data from the long term therapy with larger number of the patient cohort would be required. In this regard, the cross-regional studies would be advantageous.
Response 3: Dear reviewer, we really appreciated your question and recommendation. Majority of our study patients in both groups received guideline-direct cardioprotective therapy such as ASA and statin, because the medical therapy is the only solution for these high-risk patients with severe CAD unsuitable for any coronary intervention after heart team approach, especially for those in the control group. In order to answer your question, we need longer follow-up time and more patient number. However, in fact, patients’ enrollment is quite difficult because most patients with severe diffuse CAD can be managed with either percutaneous or surgical coronary intervention after heart-team discussion. That is why our enrollment period was long and patient’s number was limited. We will try our best for more and longer data collection. Thank you for your understanding.
Comment 4: The numbers of the dose do not match over the whole manuscript. Please see the Figure 1.
Response 4: Thank you very much for your detailed review and correction. We have amended our error of CD34+ cell dosage to 3.0 x 107 in the revised Figure 1.
Comment 5: Please describe the isolation of CD34+ cells more in details (the line 143), the EPC flow cytometric analysis/ strategy using representative dot plots. Why final 300,000 cells (the line 170) were set up as a threshold for the final analysis and which gate has been chosen?
Response 5: (1) According to your recommendation, we have provided more detailed information regarding the criticism: “the isolation of CD34+ cells more in details” on the lines 147-152 of our revised manuscript. (2) Dear reviewer, because the circulatory EPC surface markers is usually very low, and in some EPC surface makers even was lower than 0.005% (please see the Figure 2). So, we found that only 300,000 cells were adequately set up as a threshold for the accurate analysis. This method is validated by our previous report [Crit Care Med 2015,43,2117-32], i.e., reference 21 of our manuscript. (3) according to your recommendation, we have provided an illustration (i.e., supplementary Figure 1) of flow cytometry for demonstrating how to gate the EPCs in our revised manuscript.
Comment 6: Please comment on the use of the protamine and discuss about the current application after CD34+ transfusion.
Response 6: We have added the statement regarding protamine effects in the lines 216-218 in the medications section of the “Methods”. We have also discussed about the current application of CD34+ therapy in the lines 380-385 in the 6th paragraph of Discussion section.
Comment 7: How the absolute count of the EPCs (Figure 2) was calculated? Please describe in detail the entire analysis relying on some representative plots.
Response 7: Dear reviewer, the absolute count of ECPs in Figure 2 was calculated by the formula of Number of CD34+ cells = (percentage of CD34+ cells) x WBC count x 103 x peripheral-blood stem cell (PBSC) volume (mL). The calculation was according to the International Society of Hematotherapy and Grafting Engineering (ISHAGE) Guidelines. We had described this method in the lines 158-160 on our original manuscript.
Furthermore, the adequate number of EPC used in this study was calculated and had been well-established based upon the previous publication (reference 24) and our previous researches (references 21 and 22). Our phase I trial has showed high-dose (3.0 x 107 cells) stem cell therapy was effective and safe for treating severe diffuse CAD with poor LV function.
Comment 8: What is the role of these separately analyzed EPCs, please comment on that more detailed.
Response 8: The circulating EPCs were separately analyzed using flow cytometric cell markers to confirm the hematopoietic cell type. In further, the purpose we analyzed these EPCs separately before and after stem cell therapy was to evaluate the angiogenesis ability by using counting number of tubules, as well as tubular, cluster and network formations on Matrigel assay. We tried to prove the subsequent improvement in the LVEF and presenting symptoms might result from post-CD34+ cell therapy-associated increase in angiogenesis. The above statements have been added in the lines 286-287 and 330-332 in the section of “Results” to make you and readers more clearly understand why we analyzed the circulating EPCs.
Thank you very much for your kind help!
We greatly appreciate to you for your professional comments and suggestions!
Round 2
Reviewer 2 Report
Response by the Reviewer/1.: This is a satisfactory response. The new lines in the limitations section are also fine.
Response by Reviewer/2.: This is a satisfactory response. I am awaiting the further data set on that study.
Response by Reviewer/3.: This is a satisfactory response to me. However, how far is it possible to provide a multicenter study relying on the design of the present study? CAVE: It should be ASS instead of ASA. I ‘ve made a typing error.
Response by Reviewer/4. Thank you.
Response by Reviewer/5. Thank you. The response to the point #1 is satisfactory. #2. Considering the choice of the events acquired for flow cytometric analysis and the reference, it seems that the Authors based on methods used and briefly described in their publication from 2015. However, the reference (#21, 30,000 events) relies on the other original references (#12 (200,000 events acquired), #13 (no threshold at all mentioned)). Even if the choice of the 300,000 events acquired in the present study seems to me reasonable, did the investigators validated the flow cytometric analysis before (based on the reference samples for the clinics e.g.) or is it only the randomly chosen threshold? #3. Thank you. Did you try to apply a boolean gating strategy by showing the target cell subset (e.g. CD34+CD133+) cells again in the FSC/SSC dot plot as a “backgate” which is commonly used to select hematopoietic progenitor cell with blast morphology? Did they build a homogenous population as expected? Since I can see the suppl. figure for the first time in the revised manuscript, it would be interesting to see, how significant are the differences between the FC analysis performed by the Authors in the present study and after the inclusion of the additional region as above mentioned?
Response by Reviewer/6. Thank you. The response is it satisfactory.
Response by Reviewer/7. Thank you. The response is it satisfactory.
Response by Reviewer/8. Thank you. The response is it satisfactory.
Author Response
Response by Reviewer/3.: This is a satisfactory response to me. However, how far is it possible to provide a multicenter study relying on the design of the present study? CAVE: It should be ASS instead of ASA. I ‘ve made a typing error.
Response 3: Dear Reviewer, we also hope to perform multicenter study in the future. However, not only stem cell research needs to be reviewed by our government but also not every medical center in Taiwan is capable to perform the cell-based therapy. Therefore, there is still far away from multicenter trial at present. That is why our patient’s enrollment was slow.
Response by Reviewer/5. Thank you. The response to the point #1 is satisfactory. #2. Considering the choice of the events acquired for flow cytometric analysis and the reference, it seems that the Authors based on methods used and briefly described in their publication from 2015. However, the reference (#21, 30,000 events) relies on the other original references (#12 (200,000 events acquired), #13 (no threshold at all mentioned)). Even if the choice of the 300,000 events acquired in the present study seems to me reasonable, did the investigators validated the flow cytometric analysis before (based on the reference samples for the clinics e.g.) or is it only the randomly chosen threshold? #3. Thank you. Did you try to apply a boolean gating strategy by showing the target cell subset (e.g. CD34+CD133+) cells again in the FSC/SSC dot plot as a “backgate” which is commonly used to select hematopoietic progenitor cell with blast morphology? Did they build a homogenous population as expected? Since I can see the suppl. figure for the first time in the revised manuscript, it would be interesting to see, how significant are the differences between the FC analysis performed by the Authors in the present study and after the inclusion of the additional region as above mentioned?
Response 5:
(1) Thank you for your comment.
(2) As the previous reply, since circulatory EPC makers usually show very low population, therefore, we collected 300,000 cells for flow cytometric analysis. This threshold was according to our previous Phase-I study in 2015 (ref #21). In the previous Phase-I study, we make sure again that we set the same threshold (300,000 events, not 30,000 events) for cytometric analysis, which was described at page 2120 as: “Each analysis included 300,000 cells per sample.”(Crit Care Med 2015; 43:2120).
(3) Thank you for your kindly suggestion. The boolean presentation is a rational strategy to gate subpopulation. We have performed subpopulation analysis of CD34+CD133+KDR+ again and to compare between boolean and dot plot strategies (Supplemental Figure 2). Supplemental Figure 2 (D) and (H) showed the “back-gating” of CD34+CD133+KDR+ in FSC/SSC dot plot. After comparing these two FC analyses, we found that the final percentage of CD34+CD133+KDR+ was the same, suggesting that different gating strategies didn’t affect the final percentages of subpopulations in the present study.
(4) We have added the above Supplemental Figure 2 in the line 341 in the result section of our revised manuscript.
Round 3
Reviewer 2 Report
Reviewer's Response to 3: Thank you.
Reviewer's Response to 5: (2) Thank you. This was mix-up in the opened publications at that moment. If so, It would be fine to slightly change the sentence "The procedure and protocol have been described in our previous study [21]." e.g. The validation of the flow cytomeric method as well as generation of the final protocols was performed in scope of our previous Phase-I study in 2015 [21].
(3) (4) Thank you for adding of the Suppl. Figure 2. I did not ask for the differences between density and dot plot, which is only a different kind of the illustrative presentation of the FC data. Also, I think, there is no need to show the illustration between density and dot plot. In general, the Boolean strategy is a definition in the field of the flow cytometric analysis, which involves an including and/or an excluding of gates/regions towards antigen expression/size/granularity. The presentation of the backing gate as the Authors showed (in red) in (D) and (H) new plots is only descriptive but without any quantitative statement. Why the Authors stated: “The data showed that there was no difference of analytical results”. On which data set the Authors rely? If they claim it, why there is no gate included in the last plot for backgate to generate any data (events, percentage).
From the illustration depicted with the “red” homogenous population, I can only hypothesize that no substantial differences are observed (no loss of the events) when compared e.g. plot H and G. For the D vs. C seem the population to be small and dispersed, what in turn makes doubt about the statement: “we found that the final percentage of CD34+CD133+KDR+ was the same”. At this point, please re-write the description under the suppl. Figure 2, accordingly. The quality of the suppl. Figure 2 has to be improved (resolution) before any publication. Moreover, the size of the both FC figures has to be shown in the similar way, and if different software for FC analysis was used, please supplement it under the figure and in the methods part with an appropriate description, too. Please re-write accordingly the title: 3.9. Comparison of the different gating strategies for analyzing EPC surface makers (Supplemental Figure 2).
Author Response
Responses to Reviewer's Comments (Reviewer #2):
Dear Reviewer:
Thank you so much for your detailed review of our manuscript. Your constructive criticism is greatly appreciated. We have made the following point-by-point responses to comply with your honorable suggestions (Note: The revised parts of the manuscript in response to reviewer’s comments have been marked in green color).
Comments and Suggestions for Authors:
Reviewer's Response to 3: Thank you.
Reviewer's Response to 5: (2) Thank you. This was mix-up in the opened publications at that moment. If so, It would be fine to slightly change the sentence "The procedure and protocol have been described in our previous study [21]." e.g. The validation of the flow cytomeric method as well as generation of the final protocols was performed in scope of our previous Phase-I study in 2015 [21].
Response: Thank you. According to your suggestion, we have rewritten the sentence in the lines 140-141 of methods in our revised manuscript.
(3) (4) Thank you for adding of the Suppl. Figure 2. I did not ask for the differences between density and dot plot, which is only a different kind of the illustrative presentation of the FC data. Also, I think, there is no need to show the illustration between density and dot plot. In general, the Boolean strategy is a definition in the field of the flow cytometric analysis, which involves an including and/or an excluding of gates/regions towards antigen expression/size/granularity. The presentation of the backing gate as the Authors showed (in red) in (D) and (H) new plots is only descriptive but without any quantitative statement. Why the Authors stated: “The data showed that there was no difference of analytical results”. On which data set the Authors rely? If they claim it, why there is no gate included in the last plot for backgate to generate any data (events, percentage).
Response: The inappropriate sentence and result have been corrected in our revised manuscript (please refer to supplemental figure 2).
From the illustration depicted with the “red” homogenous population, I can only hypothesize that no substantial differences are observed (no loss of the events) when compared e.g. plot H and G. For the D vs. C seem the population to be small and dispersed, what in turn makes doubt about the statement: “we found that the final percentage of CD34+CD133+KDR+ was the same”. At this point, please re-write the description under the suppl. Figure 2, accordingly. The quality of the suppl. Figure 2 has to be improved (resolution) before any publication. Moreover, the size of the both FC figures has to be shown in the similar way, and if different software for FC analysis was used, please supplement it under the figure and in the methods part with an appropriate description, too. Please re-write accordingly the title: 3.9. Comparison of the different gating strategies for analyzing EPC surface makers (Supplemental Figure 2).
Response: Thank you for your reminder. According to your professional comment, the subhead title and figure legends have been rewritten again in our revised manuscript.
Thank you very much for your kind help!
We greatly appreciate to you for your professional comments and suggestions!